# A Wash of Ethyl Acetoacetate Reduces Externally Added *Salmonella enterica* on Tomatoes

**DOI:** 10.3390/antibiotics11081134

**Published:** 2022-08-21

**Authors:** Shelley M. Horne, Birgit M. Prüß

**Affiliations:** Department of Microbiological Sciences, North Dakota State University, Fargo, ND 58108, USA

**Keywords:** tomato, *Salmonella*, food-antimicrobial, wash

## Abstract

The continuously high numbers of food-borne disease outbreaks document that current intervention techniques are not yet satisfactory. This study describes a novel wash for tomatoes that can be used as part of the food processing chain and is designed to prevent contamination with serovars of *Salmonella enterica*. The wash contains ethyl acetoacetate (EAA) at a concentration of 8% in H_2_O. This wash reduced live bacterial counts (on *Salmonella* Shigella agar) of externally added *S.* Newport MDD14 by 2.3 log, counts of *S.* Typhimurium ATCC19585 by 1.5 log, and counts of *S.* Typhimurium FSL R6-0020 by 3.4 log. The naturally occurring background flora of the tomatoes was determined on plate count agar. The log reduction by EAA was 2.1. To mimic organic matter in the wash, we added 1% tomato homogenate to the 8% EAA solution. Prior to using the wash, the tomato homogenate was incubated with the EAA for 2 h. In the presence of the tomato homogenate, the log reductions were 2.4 log for *S*. Newport MDD14 and 3 log for *S*. Typhimurium FSL R6-0020. It seems like tomato homogenate did not reduce the efficacy of the EAA wash in the two *S. enterica* serovars tested. We propose the use of EAA as a wash for tomatoes to reduce bacterial counts of *S. enterica* well as naturally occurring background flora.

## 1. Introduction

In the context of healthy eating habits, produce has become more important in recent years. This raises the question of how to keep produce microbiologically safe. The Centers for Disease Control and Prevention (CDC; www.cdc.gov, accessed on 31 July 2022) list numerous outbreaks of *Salmonella enterica* in conjunction with produce, including onions (*S.* Oranienburg, 2022; *S.* Newport, 2020), pre-packaged salads (*S.* Typhimurium, 2021), peaches (*S.* Enteriditis, 2020), alfalfa sprouts (*S.* Newport, 2010; *S.* StPaul, 2009; *S.* I 4,[5],12:i:, 2011) and tomatoes (*S.* Typhimurium, 2006). In the United States, the estimated annual number of infections by *Salmonella*, caused primarily by the consumption of food runs, is approximately 1.35 million infections, 26,500 hospitalizations, and 420 deaths (CDC).

The four outbreaks of *S. enterica* in Virginia that were associated with tomatoes (*Lycopersicon esculentum*) are examples of contamination in the pre-harvest environment [1]. Among several factors that were investigated after these outbreaks, untreated pond water as irrigation water and fresh poultry litter as soil amendment increased the likelihood of *S. enterica* in tomato plots. The spread of *S. enterica* within agricultural fields can be facilitated by rain that permits the formation of *Salmonella* aerosols [2]. Post-harvest contamination with *S. enterica* can occur during the slicing of tomatoes [3] or by means of cross-contamination from meat on kitchen equipment [4].

Decontamination efforts in personal homes include rinsing tomatoes. Also a point-of-use UV appliance could be used in homes [5]. Additional physical decontamination methods include high hydrostatic pressure [6]. Among the chemical decontaminations, a brush roller and spray system that uses sodium hypochlorite (NaOCl) at 100 mg/mL reduced *S. enterica* by 4 log at 15 s of treatment, while 80 mg/mL of peroxyacetic acid (PAA) reduced *S. enterica* by 3 log, also at 15 s of treatment [7]. Simulated chlorine washes at 25 to 150 mg of free chlorine reduced *S. enterica* by 2 to 3 log in a commercial setting [8]. Altogether, it seems like there are ample decontamination methods for tomatoes, and yet outbreaks still happen. In this study, we present our evidence towards a new food anti-microbial, ethyl acetoacetate.

An early study by our own research laboratory identified ß-phenylethylamine and acetoacetate as inhibitors of growth and biofilm in liquid beef broth growth medium and refrigeration temperature [9]. Ethyl acetoacetate (EAA) is chemically and structurally similar to acetoacetate but was more effective at reducing growth, live bacterial counts, and biofilm in three bacterial pathogens [10]. It is also more cost-effective. Since its first identification as an anti-microbial, EAA has been used in one clinical application [11] and two food safety applications, where it was effective at reducing spoilage bacteria on beef [12] and chicken [13]. For beef, EAA was proposed as an additive [12]. For chicken [13] and this study on tomatoes, EAA was suggested as a food processing aid, where the residual concentration that remains on the food from washing is much smaller. According to the FDA (www.fda.gov, accessed on 31 July 2022), processing aids do not have to be included on the label but will still require FDA approval. The current approval of EAA is for use as a flavoring additive under 21CFR172.515. EAA is used under Flavis No. 9.402. Since food preparation often includes cooking, the heat stability of EAA has been determined at 165 F (73.9 °C; inside of a burger) and 190 F (87.8 °C; outside of a burger). EAA was stable chemically and as an anti-microbial up to 190 F [14]. This study describes EAA as a wash for tomatoes that reduces counts of *S. enterica* by three serovars.

## 2. Results

### 2.1. A Concentration of 8% Was Effective for EAA at Reducing the Number of S. Enterica after an Incubation of 5 Minutes

*S.* Typhimurium FSL R6-0020 and *S.* Typhimurium ATCC 19585 were grown to an OD_600_ of 1.0 and incubated for 5 min with increasing concentrations of EAA in H_2_O. The log_10_ CFU/mL of *S.* Typhimurium FSL R6-0020 decreased to 78% at a concentration of 7% EAA and reached the detection limit (2 log) at a concentration of 8% EAA (Figure 1). The log_10_ CFU/mL of *S.* Typhimurium ATCC19855 declined to 45% at a concentration of 7%. At a concentration of 8%, the number of bacteria had also reached the detection limit. It was concluded to use a concentration of 8% EAA for this study because this was the lowest concentration at which the log_10_ CFU/mL dropped to the detection limit.

### 2.2. The EAA Wash Reduced Counts of Externally Added S. enterica and Counts of Naturally Occurring Background Flora

Tomatoes were inoculated with ~10^9^ bacteria of each of three *S. enterica* serovars and consecutively washed with either H_2_O or 8% EAA in H_2_O. A control group that received PBS instead of bacteria was included. Natural occurring background flora was enumerated on Plate count agar (PCA) plates, and *S. enterica* on Shigella-*Salmonella* Agar (SSA) plates. The log_10_ CFU/g tomato data are summarized in Figure 2. The one-way ANOVA yielded a *p*-value < 0.0001, indicating that there are statistically significant differences between the groups.

For *S.* Newport MDD14 (Figure 2), there is a 2.3 ± 1.7 log reduction when comparing the *S. enterica* counts from the SSA plates between H_2_O and EAA wash (first set of bars in Figure 2). The *p*-value from the unpaired, one-tailed *t*-test that compared the CFU/g tomato data from the H_2_O and the EAA wash was 0.025, indicating the statistical significance of the difference. 

For *S.* Typhimurium ATCC 19585 (second set of bars in Figure 2), the EAA wash reduced the *S. enterica* counts from the SSA plates by 1.5 ± 1.5 log. The *p*-value from the *t*-test was 0.063, which is not indicative of the statistical significance of the difference. Since the *p*-value from the parametric *t*-test is close to the cut-off of 0.05 and the sample size is too small to calculate normality, we also used the non-parametric Wilcoxon test. The *p*-value from this test was 0.069, which is very similar to the *p*-value from the *t*-test and also not significant. When looking at the distribution of the data from the water wash, it is notable that in two of five experiments, there were few bacteria recovered already after the H_2_O wash. We believe that this serovar may have a reduced ability to attach to the tomatoes and therefore got washed off by H_2_O in some but not all of the experiments. For the three experiments that permitted the recovery of *S. enterica* from the H_2_O-washed tomatoes, we calculated a log reduction of 2.5 ± 0.8 at a *p*-value from the *t*-test of 0.053, which is just outside the significance cut-off.

For *S.* Typhimurium FSL R6-0020 (third set of bars in Figure 2), the log reduction by the presence of EAA in the wash was calculated at 3.4 ± 0.95. This was the highest log reduction in our experiments, which was accompanied by a very low *p*-value from the *t*-test of 0.0033. 

The background experiment with PBS was analyzed on PCA plates as we did not add any *S. enterica* serovars (fourth set of bars in Figure 2). Counts of naturally occurring background flora were reduced by 2.13 ± 0.95 log. The *p*-value from the *t*-test was very low at 0.0009. 

### 2.3. Recovery of Bacteria with 0.1% Silwet Leads to a Smaller Recovery of S. enterica from the Tomatoes

For the past experiment, bacteria were recovered from the tomatoes by homogenizing the whole tomatoes and plating serial dilutions onto selective agar plates. To test the possibility that recovering the bacteria from the tomatoes with detergent might lead to an increased (relative to homogenization) recovery of the inoculated *S. enterica*, we repeated the experiment with *S.* Typhimurium FSL R6-0020 and recovered bacteria with a solution of 0.1% silwet L-77. The log_10_ CFU/g data are presented in Figure 3. A one-way ANOVA yielded a *p*-value of <0.0001, indicating statistically significant differences between some of the test groups of this experiment. 

The first observation from this experiment is that the recovery of bacteria with silwet L-77 (Figure 3) is much reduced when compared to the recovery of bacteria by means of tomato homogenization (Figure 2). When bacteria were recovered by homogenization of the tomatoes, the *S. enterica* count on the SSA plates for *S.* Typhimurium FSL R6-0020 after H_2_O wash was 4.37 log_10_ CFU/g (Figure 2). When bacteria were recovered with silwet L-77, the *S. enterica* count on the SSA plates after H_2_O wash was 3.08 log_10_ CFU/g (Figure 3, second grey bar). This is a 1.3 log difference in recovery between the two methods. Moreover, the recovery of *S.* Typhimurium FSL R6-0020 was better when the bacteria were recovered from the tomatoes with H_2_O that did not contain silwet L-77. For this test group, the recovery was 3.5 log_10_ CFU/g (Figure 3, first gray bar).

As a second observation, log reductions caused by the EAA wash were smaller than those obtained from the previous experiment, where bacteria were recovered by homogenizing the tomatoes. When comparing the H_2_O-washed and the EAA-washed tomatoes, the log reduction after silwet L-77 recovery (bars 3 and 4 in Figure 3) was 1.8 ± 0.75 with a *p*-value of 0.0004, which is indicative of the statistical significance of the difference. In comparison, in the previous experiment, the respective log reduction was 3.4 ± 0.95 (Figure 2). *S. enterica* counts from the PBS samples (bars 5 and 6 in Figure 3) were below the detection limit of 14 CFU/g for both washed and a *p*-value could not be computed. It seems like recovery of bacteria by tomato homogenization is preferable over recovery with silwet L-77 with respect to both the total number of recovered *S. enterica* and log reductions in response to 8% EAA.

### 2.4. The Effectiveness of EAA Ws Not Reduced in the Presence of 1% Tomato Homogenate in the Wash

The initial experiment was repeated with *S.* Newport MDD14 (Figure 4) and *S.* Typhimurium FSL R6-0020 (Figure 5). In this experiment, the 8% EAA wash also contained 1% of tomato homogenate to test for the effect of organic load on the EAA wash. The one-way ANOVA comparing all groups yielded a *p*-value of 0.2206, but Bartlett’s correction yielded a *p*-value < 0.0001. We did perform one-tailed unpaired *t*-tests for all relevant comparisons. 

For the *S.* Newport MDD14 serovar (Figure 4), there is a reduction of 1.6 ± 1.2 log when comparing the bacterial counts from the PCA plates between the H_2_O and the EAA wash (first set of bars in Figure 4). For the *S. enterica* counts from the SSA plates (second set of bars in Figure 4), this log reduction was 2.6 ± 0.7 log. The unpaired, one-tailed *t*-test that was performed on the CFU/g tomato data was not indicative of the statistical significance of this difference in both of those comparisons. When the wash contained 1% tomato homogenate (the third set of bars in Figure 4), the bacterial counts from the PCA plates were reduced by 2.55 ± 0.65 log by the EAA wash. Comparing the CFU/g tomato data from the H_2_O and the EAA washes, the *p*-value from the parametric *t*-test was 0.039, which is indicative of the statistical significance of the difference. The *S. enterica* counts from the SSA plates (final set of bars in Figure 4) were reduced by 2.4 ± 0.7 log with a *p*-value from the *t*-test of 0.021. In summary, the EAA wash was effective in the presence of 1% tomato homogenate (log reduction on SSA of 2.4, Figure 4) to the extent that it was effective in the absence of tomato homogenate (where we calculated a log reduction of 2.3 in Figure 2 and a log reduction of 2.6 in Figure 2). 

While performing the experiments, we noticed that the tomato homogenate itself seemed to have a small effect on the resulting bacterial counts. The largest of these reductions was the total bacterial count from the PCA plates when comparing the EAA wash and the EAA/tomato wash (first and third white bars in Figure 4). The log reduction was 1.2 ± 1.2 with a *p*-value from the *t*-test of 0.1793, which is indicative of a lack of statistical significance of the difference. However, at this point, we can not exclude the possibility that the presence of the tomato homogenate in the wash itself may have a small reduction effect on the bacterial counts recovered from the tomatoes. 

The one-way ANOVA for the *S.* Typhimurium FSL R6-0020 group of samples was 0.0023, indicating statistically significant differences between some of the samples. When comparing the bacterial counts from the PCA plates between the H_2_O and the EAA wash (first set of bars in Figure 5), the log reduction was 1.74 ± 0.36 log. For the *S. enterica* counts from the SSA plates (second set of bars in Figure 5), this log reduction was 2.64 ± 0.7 log. The *t*-test that was performed on the first of these comparisons (PCA plates) was 0.0293, which is indicative of a statistical significance between the two groups. For the second of these comparisons (SSA plates), the *p*-value was 0.0427, which is also indicative of the statistical significance of the difference between the two groups. 

When the wash contained 1% tomato homogenate (the third set of bars in Figure 5), the bacterial counts from the PCA plates were reduced by 1.85 ± 1.2 log by EAA. This is slightly higher than the log reduction of 1.7 in the absence of tomato homogenate. Compared to the CFU/g tomato data from the H_2_O and the EAA washes on the PCA plates, the *p*-value from the parametric *t*-test was 0.0506, which is a borderline significance for the difference. The *S. enterica* counts from the SSA plates (final set of bars in Figure 5) were reduced by 3 ± 0.6 log with a significant *p*-value from the *t*-test of 0.0175. This log reduction is very similar to or even slightly higher than that from the experiment that did not include tomato homogenate (2.6 log, Figure 5). In summary, the EAA wash had an effectiveness in reducing externally added *S.* Typhimurium R6-0020 in the presence of 1% tomato homogenate that was similar to or even higher than that of EAA (3 log vs. 2.8 log) at the same concentration of 8% EAA but in the absence of tomato homogenate.

## 3. Discussion

The current study describes a novel wash for tomatoes to prevent contamination with *S. enterica.* At a concentration of 8%, a wash of EAA reduced live bacterial counts (on SSA plates) of externally added *S.* Newport MDD14 by 2.3 log (Figure 2). Counts of *S.* Typhimurium ATCC19585 were reduced by 1.5 log (Figure 2), while counts of *S.* Typhimurium FSL R6-0020 were reduced by the EAA wash by 3.4 log (Figure 2). Naturally occurring background microorganisms from PCA plates were reduced by 2.1 log (Figure 2). An alternative method of recovering bacteria from tomatoes yielded no improvement in either the total number of recovered bacteria or log reductions (Figure 3). To determine whether the efficacy of EAA would withstand tomato exudates, the experiment was repeated in the presence of 1% tomato homogenate in the 8% EAA wash (Figure 4 and Figure 5). The tomato homogenate had been incubated with the EAA for 2 h. In the presence of the tomato homogenate, the log reductions were 2.4 log for *S.* Newport MDD14 (vs. 2.6 log in the same experiment, Figure 4) and 3.1 for *S.* Typhimurium FSL R6-0020 (vs. 2.64 from the same experiment, Figure 5). It seems like tomato homogenate did not reduce the efficacy of the EAA wash in the two *S. enterica* serovars tested. We propose the use of EAA as a wash for tomatoes to reduce bacterial counts of *S. enterica*.

As a note on the inconsistent data that were obtained with *S.* Typhimurium ATCC19585, this is a serovar of the LT2 clade that carries a defect within the *rpoS* gene [15]. The defect renders the entire LT2 clade avirulent [15]. In addition, *rpoS* is a positive regulator of the curli operon, which is involved in attachment [16,17]. In agreement with this, an *rpoS* mutant exhibited a reduced ability to attach to alfalfa plant tissue [18]. The reason for the lack of reproducibility of our data with this serovar is that in two of the five experiments, the bacteria got washed off the tomatoes with H_2_O already and, hence, the addition of EAA to the wash could not cause any further reduction. We believe the reduced ability of this specific serovar of *S. enterica* to attach could be the cause of the occasional removal of the bacteria from the tomato surface by pure H_2_O. In the three experiments where we were able to retrieve bacteria after the H_2_O wash, the log reduction caused by EAA was 2.5. This is within the range of log reductions we obtained from the other two serovars. We conclude that EAA was effective against all three of our tested *S. enterica* serovars.

In order to assess the usefulness of EAA relative to existing technologies that aim to reduce *S. enterica* on tomatoes, one has to compare the log reductions obtained in this study with those obtained from currently used anti-microbial intervention techniques. Examples of such techniques are summarized in Table 1. 

A commonly used chemical that is added to the tomato wash is chlorine, which is most active as free chlorine or hypochlorite (HOCl). In two independent studies, free chlorine was incubated with the tomatoes for 1 and 2 min, resulting in log reductions of between 2 and 3 log [8,19]. This is similar to our log reductions for *S.* Newport MDD14 and *S.* Typhimurium FSL R6-0020. Problems with chlorine washes for produce include the consumption of free chlorine that is necessary to reduce bacterial numbers by fresh-cut produce exudates [27] and the production of stronger biofilm by the bacteria upon chlorine stress via induction of the oxidative stress response [28]. In tomatoes, it has been shown that field debris and defective tomatoes at less than 1% make up for 55.5% of the chlorine demand in the wash [29]. In our study, the addition of 1% tomato homogenate to the 8% EAA wash did not reduce the efficacy of the wash, which is an advantage of EAA overusing chlorine. In addition, EAA was identified as an inhibitor of biofilm on polystyrene plates by three pathogens [10] and an inhibitor of biofilm in silicone tubing by several pathogens, including *S. enterica* [11]. This is the second advantage of EAA over chlorine that induced the formation of biofilm [28]. EAA does have FDA approval as a flavoring agent under 21CFR172.515 and is used under Flavis No. 9.402.

The remaining anti-microbial intervention techniques that aim at reducing *S. enterica* on tomatoes and are listed in Table 1 include chlorine dioxide (ClO_2_), peroxyacetic acid (PAA), numerous other acids, UV light, and phages. For the latter [26], the incubation time was very long, and the study may not be very suitable to be compared to ours; the concept may be interesting, however. For the remaining technologies, the log reductions are all in the range of the ones that we determined for *S.* Newport MDD14 and *S.* Typhimurium FSL R6-0020 or smaller than those. 

The experiment in Figure 2 also included a comparison of naturally occurring background flora as enumerated on PCA plates and included many of the naturally found spoilage microorganisms and pathogenic bacteria in the presence and absence of EAA in the wash water. Spoilage bacteria on tomatoes include many molds, fungi, and yeast. Specific examples include *Alternaria alternata* and *Stemphylium vesicarium*, against which ClO_2_ has been successfully used as well [30]. Similar to that described above for *S. enterica*, the anti-microbial activity against yeast and molds of some of the sanitizers (e.g., sulfuric acid) can be reduced by organic load (e.g., tomato debris) in the wash. Bacterial background flora can include *Pseudomonas syringae* pv tomato, which causes bacterial speck disease on tomatoes [31,32]. *Pseudomonas floridensis* is a novel bacterial pathogen that has been isolated from tomatoes just recently. 

In our study, EAA reduced the live bacterial counts of naturally occurring microflora on the tomatoes by 2.1 log (Figure 2). This log reduction was obtained from PCA plates, which are a non-selective growth medium that permits the growth of many microorganisms under the indicated temperature and under aerobic conditions. Since *Pseudomonas* can sometimes be found on tomatoes (see above), we enumerated bacteria on PSA (pseudomonas agar base) plates as part of the same experiment (data not shown). In three of the five experiments, we were unable to detect *Pseudomonads* in even the undiluted samples (lower limit of detection = 49 CFU/g). In the remaining two experiments, the log reductions were 1.6 and 1.2 log. We conclude that *Pseudomonads* were either not very prevalent on our tomatoes or got washed off already with pure water. In cases of presence, EAA was able to reduce the number of *Pseudomonads* by 1.4 ± 0.2 log. Altogether, the effectiveness of 8% EAA in the wash water was similar or slightly less for the naturally occurring background flora than it was against the externally added *S. enterica* serovars. This is different from using EAA on chicken or beef, where we observed a strong effect against the background flora but a much lesser effect against externally added pathogens [12,13].

In conclusion, we believe that it will be a worthwhile endeavor to include EAA on the list of antimicrobials that are worth future research, in particular in a setting that is closer to an actual food processing chain.

## 4. Materials and Methods

### 4.1. Bacterial Serovars and Growth Conditions

Three serovars of *S. enterica* were used in this study (Table 2). *S.* Typhimurium FSL R6-0020 is also designated TB0041 [33]. *S.* Typhimurium ATCC 19585 (ex Kauffmann and Edwards) Le Minor and Popoff is also designated LT2 [34,35,36] and was obtained from the American Type Culture Collection (ATCC; www.atcc.org, accessed on 25 October 2021). *S.* Newport MDD14 is also designated TB0581 [37]. For the overnight cultures, bacteria were grown at 34 °C in Brain Heart Infusion (Difco^TM^ BHI; BD Life Sciences, Gurgaon, India).

### 4.2. Determine the Effective Concentration of EAA against Salmonella

Overnight cultures of *S.* Typhimurium FSL R6-0020 and *S.* Typhimurium ATCC 19585 were diluted 1:10 into BHI, grown at 34 °C for 2 h, and adjusted to an OD_600_ of 1.0, at which time cultures contained approximately 10^8^ to 10^9^ colony forming units (CFU)/mL. Solutions of 0%, 5%, 6%, 7%, 8%, 9%, and 10% EAA (Alfa Aesar, Ward Hill, MA, USA) were produced in H_2_O. In total, 1 mL of bacterial culture was added to 9 mL of EAA at the indicated concentrations. The solutions were incubated for 5 min. Serial dilutions were produced, 10 µL of each dilution was plated onto *Salmonella* Shigella Agar (BBL^TM^ SSA; BD Life Sciences, St. Louis, MO, USA).

### 4.3. Performance of the Tomato Experiment

Overnight cultures of each of the three *S. enterica* serovars were diluted 1:10 in BHI and grown for 2 h at 34 °C. Cultures were adjusted to an OD_600_ of 1.0. To concentrate the bacteria, these cultures were centrifuged and resuspended in 100 µL PBS. The starting concentrations of the bacteria were determined by plating serial dilutions onto SSA plates. Inocula contained ~10^9^ CFU.

Unwaxed, organic grape tomatoes were purchased from a local grocery store. Tomatoes were weighed before use. For each wash, two tomatoes were spot inoculated in modification of a previously described protocol [38]. Briefly, tomatoes were located on weighing dishes during the inoculation with 100 µL of a culture containing approximately 10^9^ CFU of the respective *S. enterica* serovar in PBS. For each negative control, two tomatoes received 100 µL of PBS without bacteria. Tomatoes were dried for 2 h under the BSL2 biosafety cabinet with airflow. EAA washes were performed at a concentration of 8% EAA in H_2_O, the control wash was H_2_O. The two tomatoes that had been inoculated with the same *S. enterica* serovar (or negative control) were transferred into the same Ziploc bag, where they received 10 mL of wash (EAA or H_2_O) and were incubated for 5 min at room temperature. For 1 min, tomatoes were agitated in the Ziploc bags. Tomatoes were dried for 10 min on sterile drying racks.

### 4.4. Production of the Tomato Homogenate

The tomato experiment was repeated in the absence and presence of 1% tomato homogenate in the H_2_O or 8% EAA wash. Four to six grape tomatoes were transferred into a stomacher bag and homogenized in the Seward Stomacher for 30 s at 230 rpm. This solution was directly used as homogenate and added to the EAA washes at a concentration of 1%.

### 4.5. Recovery of Bacteria from the Tomatoes 

Bacteria were recovered from the tomatoes by means of homogenization. In one experiment, bacteria were recovered by means of 0.1% silwet L-77.

#### 4.5.1. Recovery by Eans of Homogenization

Tomatoes that had received identical treatment combinations (*S. enterica* or BPS/EAA or H_2_O) were transferred into a stomacher bag. In total, 10 mL of Maximum Recovery Diluent (MRD; Millipore Sigma, St. Louis, MO, USA) was pipetted into the bag. Tomatoes were homogenized in the Seward Stomacher 400 Circulator (Cole Parmer, Vernon Hills, IL, USA) stomacher for 30 s at 230 rpm. Homogenates were removed from the bags and serially diluted at steps of 1:10. Of each dilution, 100 µL were plated onto plate count agar (Difco PCA; BD Life Sciences, St. Louis, MO, USA) to determine the aerobic bacterial count and SSA to determine the *S. enterica* count. 

#### 4.5.2. Recovery of Bacteria from the Tomatoes by Means of 0.1% silwet L-77 wash

One tomato was transferred into a new Ziploc bag and supplemented with 5 mL of 0.1% silwet L-77 (PlantMedia, Dublin, Ireland) in MRD. Tomatoes were agitated in the bags for 2 min and incubated for another 3 min. In total, 1 mL of the solution was removed, and serial dilutions were produced at steps of 1:10. Dilutions were plated onto PCA and SSA plates as described in the homogenization paragraph (4.4.1).

### 4.6. Enumeration of Bacteria

Naturally occurring background flora were calculated from the PCA plates, *S. enterica* counts were taken from the SSA plates. Colonies from each of the dilutions were counted and converted to CFU/mL, CFU/g tomato, and log_10_ CFU/g tomato. Log reductions were calculated for each of the comparisons, using the log_10_ (a/b) formula, where ‘a’ is the CFU/g tomato count from the H_2_O control wash and ‘b’ the CFU/g tomato count from the EAA wash. Each experiment was performed in 4 to 5 biological replicates. CFU/mL, CFU/g tomato, log_10_ CFU/g tomato, and log reductions were calculated separately for each replicate. Averages and standard deviations were calculated for log_10_ CFU/g tomato and log reduction data.

Statistical analysis was started with a one-way ANOVA and followed with the parametric, unpaired, one-tailed *t*-test. Both the ANOVA and the *t*-test were performed using GraphPad Prism 9 (GraphPad Software, version 9.4.1, San Diego, CA, USA). Comparisons that were calculated include the H_2_O and EAA washes for each of the serovars (Figure 2, Figure 3, Figure 4 and Figure 5) or absence or presence of tomato homogenate (Figure 4). For one comparison, the non-parametric Wilcoxon test was used in addition to the *t*-test. This was performed in Visual Basics through Excel. For both, the *t*-test and Wilcoxon, a *p*-value of 0.05 was used as a cut-off to define the statistical significance of the respective difference.

## 5. Patents

This work did not result in any patents. However, there is a patent pending from research that led to the question addressed by this study. The patent application is listed as US 2019/0082688, published on 21 March 2019.

## Figures and Tables

**Figure 1 antibiotics-11-01134-f001:**
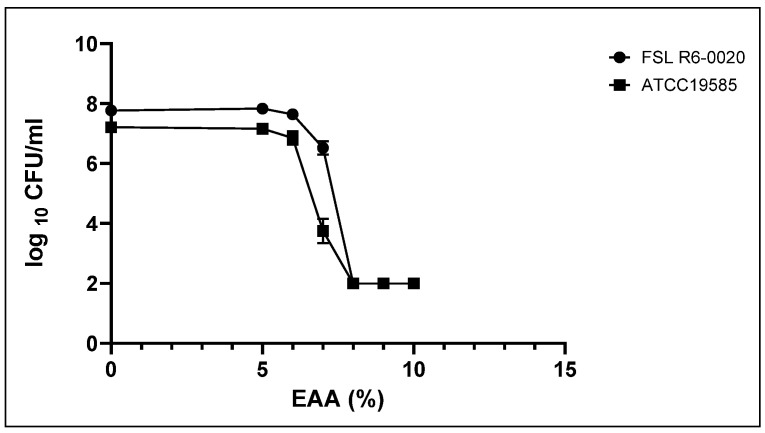
Concentration dependence of the survival of *Salmonella* after 5 min of incubation with EAA. Data are expressed in CFU/mL from three replicate experiments.

**Figure 2 antibiotics-11-01134-f002:**
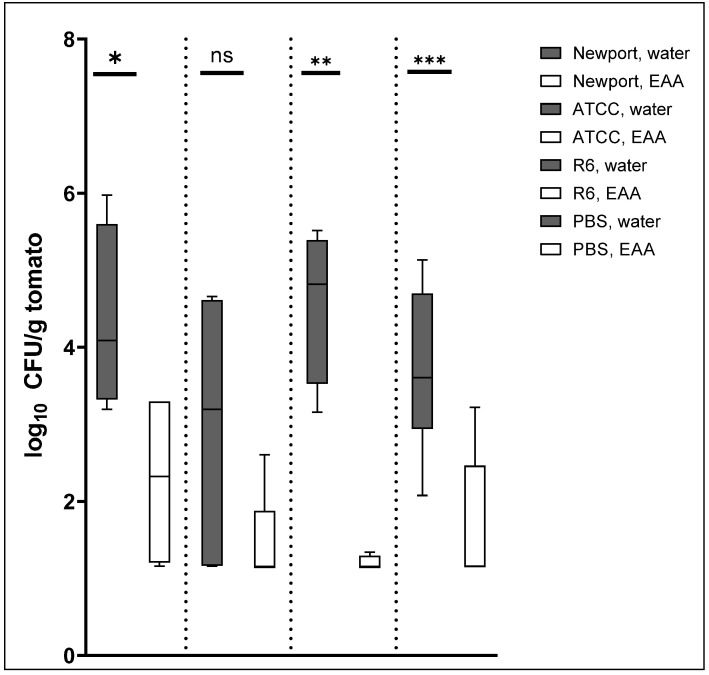
Effect of EAA on externally added *S. enterica* and naturally occurring background flora. Newport, *S.* Newport MDD14; ATCC, *S.* Typhimurium ATCC 19585; R6, *S.* Typhimurium FSL R6-0020. The log_10_ CFU/g tomato data are plotted for each of the samples (grey bars, H_2_O white bars, EAA). The *S. enterica* counts were taken from the SSA plates, the counts for naturally occurring background flora from the PBS tomatoes were taken from the PCA plates. The asterisks are indicative of the significance of the *p*-values from the *t*-test in ranked order (*** 0.0009, ** 0.0033, * 0.025, ns not significant).

**Figure 3 antibiotics-11-01134-f003:**
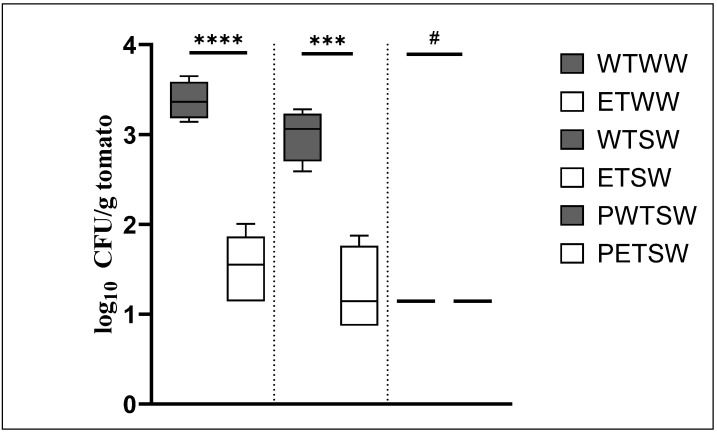
Recovery of *S.* Typhimurium FSL R6-0020 from the tomatoes by means of silwet L-77. Log reductions were calculated between the 8% EAA wash and the H_2_O wash. The log_10_ CFU/g tomato data are plotted for each of the samples (grey bars, H_2_O; white bars, EAA). The asterisks are indicative of the significance of the *p*-values from the *t*-tests (**** < 0.001, *** 0.0004, # not applicable as both groups yield identical data). WTWW, FSL R6-0020 with H_2_O wash and H_2_O recovery; ETWW, FSL R6-0020 with EAA wash and H_2_O recovery; WTSW, FSL R6-0020 with H_2_O wash and silwet L-77 recovery; ETSW, FSL R6-0020 with EAA wash and silwet L-77 recovery; PWTSW, PBS with H_2_O wash and silwet L-77 recovery; PETSW, PBS with EAA wash and silwet L-77 recovery.

**Figure 4 antibiotics-11-01134-f004:**
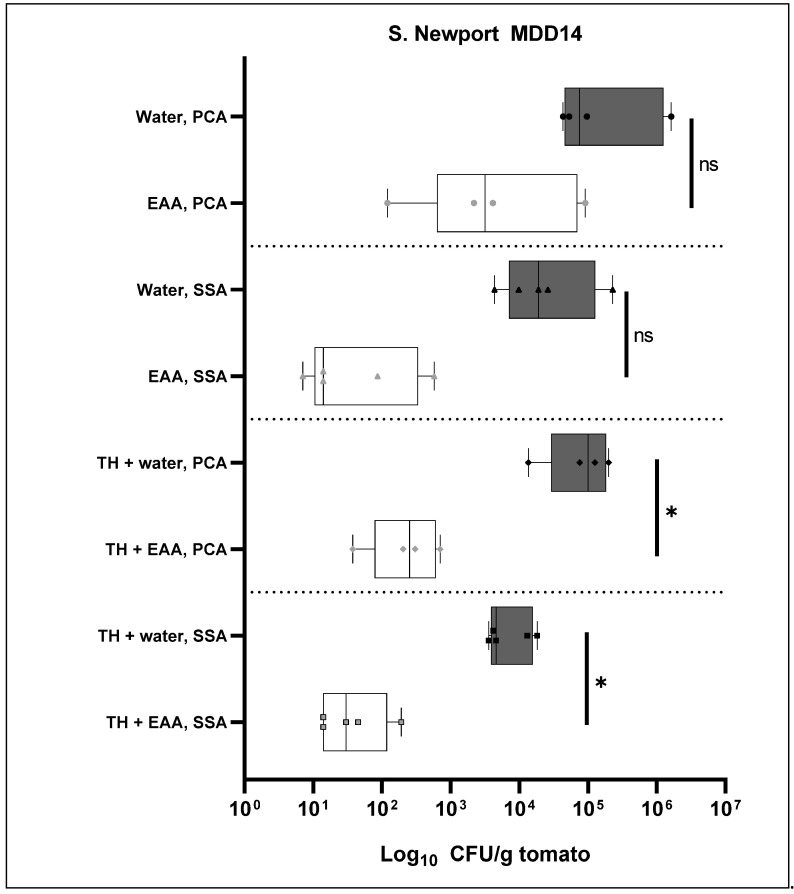
Effect of % tomato homogenate in the 8% EAA wash for the *S.* Newport MDD14 serovar. The log_10_ CFU/g tomato data are plotted for each of the samples (grey bars, H_2_O; white bars, EAA). The asterisks are indicative of the significance of the *p*-values from the *t*-test (* 0.039 and 0.021, ns not significant). TH, 1% tomato homogenate was added to the 8% EAA solution or H_2_O.

**Figure 5 antibiotics-11-01134-f005:**
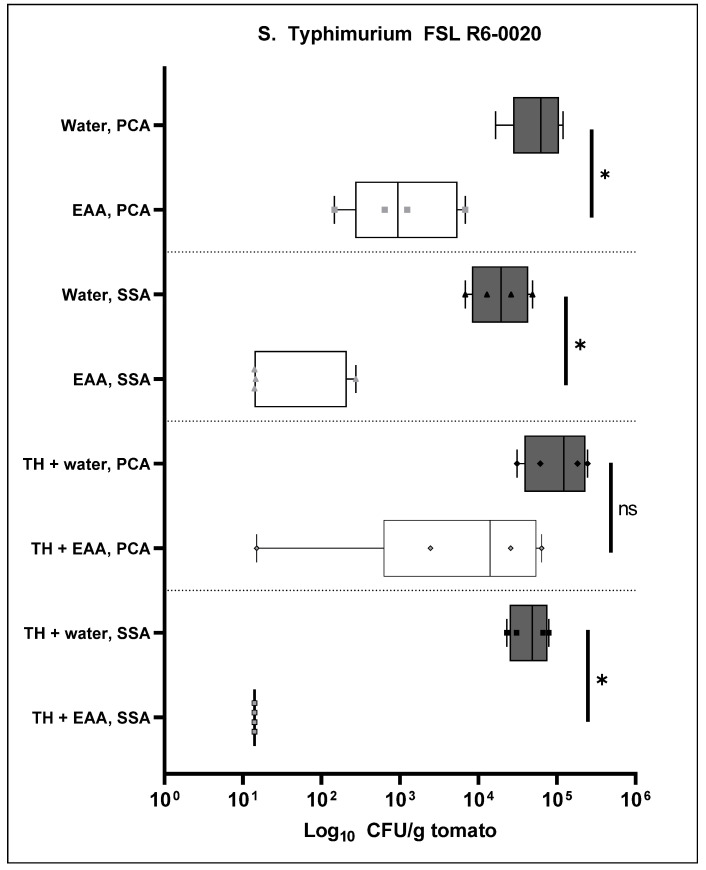
Effect of 1% tomato homogenate in the 8% EAA wash for the and *S.* Typhimurium FSL R6-0020 serovar. The log_10_ CFU/g tomato data are plotted for each of the samples (grey bars, H_2_O; white bars, EAA). The asterisks are indicative of the significance of the *p*-values from the *t*-tests (* 0.0293, 0.0426, 0.0175, ns not significant). TH, 1% tomato homogenate was added to the 8% EAA solution or H_2_O.

**Table 1 antibiotics-11-01134-t001:** Examples of currently used anti-microbials to reduce *S. enterica* on tomatoes.

Anti-Microbial	Concentration	Incubation Time	Log Reduction	Reference
EAA	8%	5 min	1.5 to 3.4	This study
Chlorine-simulated wash	25 to 150 mg/L free chlorine	1 min	2 to 3 log	[8]
Chlorine	8 ppm free chlorine	2 min	2.6 log	[19]
ClO_2_	~12 g/m^3^	30 min	~3 log at 15 °C	[20]
Peroxyacetic acid (PAA)	8 ppm free chlorine plus 1% PAA	2 min	~1.7 log at 25 °C	[19]
PAA plus sulfuric acid	90 ppm	60 s	1.75 log	[21]
Varies organic acids	0.5%	1 min	2.7 log	[22]
Pelargonic acid	0.5%	0 h	>1 log	[23]
Nisin based organic acid	10%	5 to 30 min	~ 2 log	[24]
Water-assisted UV (WAV)	NA		3.6 log	[25]
*Mycoviridae/Syphoviridae*	cocktail	3 days	>4 log	[26]

**Table 2 antibiotics-11-01134-t002:** Bacterial serovars for our study.

Bacterial Serovar	Alternative Designation	ATCC ID	Reference
*S.* Typhimurium FSL R6-0020	TB0041	Not deposited	www.foodmicrobe-tracker.com, accessed on 25 August 2021 [33]
*S.* Typhimurium LT2 (ATCC 19585)	LT2	ATCC 19585	[34,35,36]
S. Newport MDD14	TB0581	Not deposited	[37]

## Data Availability

The data presented in this study are available on request from the corresponding author.

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
