# Peer review of "A Wash of Ethyl Acetoacetate Reduces Externally Added Salmonella enterica on Tomatoes"

_antibiotics, 2022, doi:10.3390/antibiotics11081134_

Round 1

Reviewer 1 Report

Dear authors,

Thank you for the opportunity to review the paper. This one has potential for publication, but some adjustments are necessary for a better understanding of the methodology and reading.

Yours sincerely,

Introduction:

- Standardize the nomenclature of Salmonella serovars and check the text of the genus in italics. The serovar is not necessary.

line 31 and 32 - Salmonella Typhimurium, Salmonella Enteritidis

line 40 - Salmonella in italic

In the first paragraph, it is better to put the references, as the reports are published on the CDC website.

line 62 and 63 - enter the temperature in °C too.

In the introduction you have already described some results of the study, I think it is not necessary.

Materials and Methods:

Acknowledgments can be placed in acknowledgments to Dr. Teresa Bergholz for making the Salmonella strains available.

Did you have negative control? Without Salmonella inoculation and EAA Wash. It wasn't clear for me. Including the PCA count to clarify the mesophilic bacterial count of tomatoes.

The supplementary file with UFC/mL could be made available. 

line 369-371 - The strain S. enterica subsp. enterica (ex 369 Kauffmann and Edwards) Le Minor and Popoff serovar Typhimurium is also designated 370 LT2 and was obtained from the American Type Culture Collection (ATCC; 371 www.atcc.org), should be described as S. enterica subsp. enterica serovar Typhimurium LT2 (ATCC 19585) only.

line 374 to 375 - it is not necessary to describe the composition of the BHI, just the brand.

line 385 to 388 - it is not necessary to describe the composition of SSA, just put the brand and the supplement of nalidix acid.

If you used a known methodology followed to determine the concentration of EAA in vitro the reference must be made. 

Why did leave the samples overnight to grow at 34ºC for two hours?

Why choose 1 ml of bacterial culture in 9 ml of EAA?

line 417 - just MRD and bland.

line 420 - PCA and bland juts.

Results: 

What are the results of the other concentrations? Why did you choose 8%?

I'm sorry, but I didn't understand item 2.3, explain in Material and methods better.

In the discussion, it would be interesting to comment on whether this concentration used is viable for use.

Why was Pseudomonas researched?

References: put the number.

Author Response

Reviewer 1 (changes are highlighted in yellow):

Dear authors,

Thank you for the opportunity to review the paper. This one has potential for publication, but some adjustments are necessary for a better understanding of the methodology and reading.

Yours sincerely,

Introduction:

- Standardize the nomenclature of Salmonella serovars and check the text of the genus in italics. The serovar is not necessary.

I think I get it. I was trying to use the long form of the names and got mixed up. I am using short forms now. The three strains are now designated S. Newport MDD14, S. Typhimurium ATCC19585, and S. Typhimurium FSL R6-0200. We are using abbreviations of these designations for the figures. Also, when we talk about Salmonella in general, we added enterica. I appreciate the reviewer bringing this to our attention. Oh, Salmonella strains was changed to Salmonella serovars. I tried to highlight all these changes, but may have missed a few.

line 31 and 32 - Salmonella Typhimurium, Salmonella Enteritidis

Done. Now line 32.

line 40 - Salmonella in italic

This has changes to S. enterica and is in italics now. Now line 41.

In the first paragraph, it is better to put the references, as the reports are published on the CDC website.

I could not find the references on the CDC website. Nor on PubMed. There are links to the FDA website, but that are not peer-reviewed journal articles either.

line 62 and 63 - enter the temperature in °C too.

Done. This is on lines 68 and 69 now.

In the introduction you have already described some results of the study, I think it is not necessary.

Agreed. The paragraph was shortened to a single sentence at the end of the previous paragraph. Lines 70 and 71.

 Materials and Methods:

Acknowledgments can be placed in acknowledgments to Dr. Teresa Bergholz for making the Salmonella strains available.

Agreed. The reference to Dr. Bergholz was deleted. This is the paragraph from lines 331 to 337. The deletions are not highlighted.

Did you have negative control? Without Salmonella inoculation and EAA Wash. It wasn't clear for me. Including the PCA count to clarify the mesophilic bacterial count of tomatoes.

Yes, we do have counts for PBS (no bacteria) and both H2O and EAA wash. These are columns 7 and 8 in Figure 2. Column 7 is the count for the mesophilic bacteria on the tomatoes without inoculation and without EAA. Column 8 is the count for the mesophilic bacteria without inoculation and with EAA.

The supplementary file with UFC/mL could be made available. 

I thought about this when writing the manuscript. I had a choice between presenting the data on the figures as CFU (or log CFU) data or as log reductions. If I had chosen to present the log reductions, I would see value in presenting the CFU data as supplement. However, I chose to present the CFU data already in the figures, which is equivalent to the raw data. Then, the log reductions are still in the text. I personally don’t see value in providing more CFU data. If the reviewer still likes to see them, I would appreciate precise instructions as to what to put into these files.

line 369-371 - The strain S. enterica subsp. enterica (ex 369 Kauffmann and Edwards) Le Minor and Popoff serovar Typhimurium is also designated 370 LT2 and was obtained from the American Type Culture Collection (ATCC; 371 www.atcc.org), should be described as S. enterica subsp. enterica serovar Typhimurium LT2 (ATCC 19585) only.

We use the designation S. Typhimurium ATCC19585 throughout the manuscript and S. Typhimurium LT2 (ATCC19585) in the Table. I believe I copied the previous designation straight from the ATCC website and did not think about it. While doing this, I found a couple of spots where I ]had written ATCC 19858. I fixed the numbers throughout the manuscript. The serovar paragraph at the beginning of the methods is from line 331 to 337 now.

line 374 to 375 - it is not necessary to describe the composition of the BHI, just the brand.

We deleted the composition and added the brand. This is line 337 now.

line 385 to 388 - it is not necessary to describe the composition of SSA, just put the brand and the supplement of nalidix acid.

 Same here. This is lines 347 and 348 now.

If you used a known methodology followed to determine the concentration of EAA in vitro the reference must be made. 

We did not follow a known methodology. We did not change anything here.

Why did leave the samples overnight to grow at 34ºC for two hours?

We wanted cultures in an active growing phase and not stationary phase bacteria. We did not change anything here either

Why choose 1 ml of bacterial culture in 9 ml of EAA?

I asked Shelley this question and this is her answer: I believe that they are asking why we used 1 ml of culture with 9 ml EAA when we were testing for the concentration to use in the experiments.  I had adjusted the concentrations of the bacteria to ODA600=1.0 and thus got ODA600=0.1 in the 10 ml test mixes.  I made my EAA solutions so that the concentrations would be 5% - 10% when 1 ml of culture was added to them.  So I did it this way to make the math easy.

line 417 - just MRD and bland.

Done. This is line 379 now.

line 420 - PCA and bland juts.

 Done. This is line 382 now.

Results: 

What are the results of the other concentrations? Why did you choose 8%?

The results for the other concentrations are in Figure 1. We now include a statement in the text that spells out the percentages to which the CFU/ml were reduced at 7%. We chose 8% because that was the lowest concentration at which the CFU/ml had dropped to the lower limit of detection. This causality was included in the text. Paragraph from lines 76 to 83.

I'm sorry, but I didn't understand item 2.3, explain in Material and methods better.

2.3. is an alternative recovery method to retrieve the inoculum Salmonella back from the tomatoes. We wanted to make sure the differences we see are not merely a consequence of incomplete recovery of the bacteria. The methods paragraph on recovery is now split into 4.4.1 and 4.4.2. I think that should help. This is lines 373 to 384.

In the discussion, it would be interesting to comment on whether this concentration used is viable for use.

This is a thought I had a while ago. I did actually contact the FDA on the 21CFR172.515 approval as flavoring additive. The approval is not very specific about the concentration, it merely says to use the lowest concentration that has the desired effect. The person I talked to on the phone did not see any problems with our concentrations. However, a phone conversation is difficult to reference, particularly without a name. In Table 1, nisin is used at 10%. So, there are examples of high concentrations of something being used. EAA is not very expensive, that should not be a problem. And after the rest of the food processing chain, there won’t be much left anyways. Seeing as I don’t have any references or own data on this topic, I would prefer not to discuss it in the paper. 

Why was Pseudomonas researched?

Pseudomonas can be part of the background flora on tomatoes, these are references 31 and 32. We referred back to this sentence when we start talking about PSA plates. Lines 314 and 315.

References: put the number.

I was aware of this, but somehow Endnote would not let me do it. I have done it now by hand right before converting to pdf. I realize I may have to do it again.

Reviewer 2 Report

This is an interesting study on the use of ethyl acetoacetate to decontaminate Salmonella during tomato washing. A 5-min washing in 8% EAA could reduce from 1.5 to 3.4 log reduction for Salmonella and 2.1 log reduction for background flora. This study demonstrates the decontamination efficiency of EAA for Salmonella during tomato washing. The manuscript could be published after proper revision. 

1.     Line 56. The authors introduced two applications of ethyl acetoacetate as an antimicrobial agent for food like beef and chicken. Please consider to briefly describe if there is any concern about chemical residues of EAA as a washing step for fresh produce in the introduction or discussion part.

2.     Figure 1. The author may consider using Log CFU/ml to keep consistency, since they uses log CFU/ml in the abstract and other paragraphs. Y-axis should be population. CFU/ml is the unit.

3.     Line 88. ~109 CFU/ml.

4.     Line 367. Please add the isolate source or related outbreaks for the strains.

5.     Table 1. The author may use consistent unit to better describe and compare the concentration. Please consider to add EAA from this study to the Table.

Author Response

Reviewer 2 (cyan highlights):

This is an interesting study on the use of ethyl acetoacetate to decontaminate Salmonella during tomato washing. A 5-min washing in 8% EAA could reduce from 1.5 to 3.4 log reduction for Salmonella and 2.1 log reduction for background flora. This study demonstrates the decontamination efficiency of EAA for Salmonella during tomato washing. The manuscript could be published after proper revision. 

  1. Line 56. The authors introduced two applications of ethyl acetoacetate as an antimicrobial agent for food like beef and chicken. Please consider to briefly describe if there is any concern about chemical residues of EAA as a washing step for fresh produce in the introduction or discussion part.

A few sentences have been added to the Introduction, differentiating food additives from food processing aids and referring to the FDA approval of EAA as flavoring additive.

  1. Figure 1. The author may consider using Log CFU/ml to keep consistency, since they uses log CFU/ml in the abstract and other paragraphs. Y-axis should be population. CFU/ml is the unit.

Good point. I did this.

  1. Line 88. ~109CFU/ml.

Actually, this was a total of 109 bacteria or CFU. Not per ml. I changed this to 109 CFU.

  1. Line 367. Please add the isolate source or related outbreaks for the strains.

The two serovars TB0041 and TB0581 were provided to me by Teresa Bergholz together with the references. These are the only references I have for these two strains. They were already in Table 2, but I have also included them in the text now. For the LT2 type strain, I went back to ATCC and found more references Among the curated references, there was a paper from the 50s by Joshua Lederberg. I found that pretty cool. There are now two more references for this strain.

  1. Table 1. The author may use consistent unit to better describe and compare the concentration. Please consider to add EAA from this study to the Table.

We already had that problem when one of my graduate students gave a presentation. I decided to use the concentrations the way they were presented in the respective papers. If I try to convert them, I may introduce errors. The molecular weight depends on what precisely they used. I would not trust my own judgement there. I did add EAA to the Table.

Round 2

Reviewer 1 Report

Dear authors,

The issues have been clarified and I believe the changes you made made the article better.

Thanks for the additional information.

PS.: Maybe the references have to change the style (numbers in the text).

Best regards.